# *Spodoptera exigua* (Hubner) (Lepidoptera: Noctuidae) Fitness and Resistance Stability to Diamide and Pyrethroid Insecticides in the United States

**DOI:** 10.3390/insects13040365

**Published:** 2022-04-08

**Authors:** Marcelo M. Rabelo, Izailda B. Santos, Silvana V. Paula-Moraes

**Affiliations:** West Florida Research and Education Center, Department of Entomology and Nematology, University of Florida, Jay, FL 32565, USA; barbosad.izailda@ufl.edu (I.B.S.); paula.moraes@ufl.edu (S.V.P.-M.)

**Keywords:** chlorantraniliprole, bifenthrin, beet armyworm

## Abstract

**Simple Summary:**

*Spodoptera exigua* is a polyphagous pest, commonly known as beet armyworm. This pest is distributed worldwide and causes yield reduction in a variety of crops. Chemical control using synthetic insecticides is the primary strategy to manage beet armyworm. In the United States, beet armyworm resistance to both chlorantraniliprole and bifenthrin insecticides was first reported in 2020. Here we examined beet armyworm fitness and stability of resistance to chlorantraniliprole and pyrethroid insecticides, since knowledge of the stability of resistance is a crucial aspect when recommending rotation of insecticides with different mode of action. Our results have indicated no decrease in bifenthrin resistance for at least a three-year period (i.e., 27 generations) when insecticide exposure was suspended. However, susceptibility to chlorantraniliprole dropped approximately 160-fold through this three-year period. Our results indicate that beet armyworm resistance to bifenthrin is stable, but unstable to chlorantraniliprole. Unstable resistance can be successfully managed at field level by switching off the selection pressure with replacement of the insecticide other than a pyrethroid.

**Abstract:**

In the United States, beet armyworm resistance to both chlorantraniliprole and bifenthrin insecticides was first reported in 2020. Here we examined beet armyworm fitness and stability of resistance to chlorantraniliprole and pyrethroid insecticides since knowledge of the stability of resistance is a crucial aspect when recommending rotation of insecticides with different mode of action. Concentration-mortality bioassays were performed with field and laboratory susceptible populations. The F2, F13, and F27 generations of the field-derived population, maintained in the laboratory without insecticide, were exposed to commercial formulations of bifenthrin and chlorantraniliprole using the leaf-dip bioassay method (IRAC n. 007). Insects from F27 had the fitness components (survival, body weight, development time) documented and compared by LSM in each insecticide concentration tested. The resistance ratio to chlorantraniliprole reached 629, 80, 15-fold at F2, F13, and F27, respectively. These results contrast with an over 1000-fold resistance ratio to bifenthrin in all generations. The field-derived population had fitness reduced by chlorantraniliprole, but not by bifenthrin. In summary, the resistance of beet armyworm to bifenthrin was stable with no shift in fitness. In contrast, resistance to chlorantraniliprole was not stable through the generations kept in the laboratory without selection pressure, likely due to fitness cost.

## 1. Introduction

*Spodoptera exigua* (Hübner) (Lepidoptera: Noctuidae) is a polyphagous pest, commonly known as beet armyworm. This pest is distributed worldwide and causes yield reduction by damaging leaves and fruits of a broad range of host plants, including vegetable, field, and flower crops [1,2]. Prior to the broad adoption of transgenic Bt cotton expressing toxins of the bacteria *Bacillus thuringiensis*, beet armyworm was a major cotton pest in the US. Currently, this pest still damages other non-Bt host plants, such as soybean, peanut, tomato, and other vegetables [3,4,5], and chemical control is the primary strategy to manage beet armyworm in these crops. Even when beet armyworm occurs as a secondary pest, frequent insecticide application targeting other pests in high-value commodity crops, such as peanut, intensifies beet armyworm insecticide exposure [5,6,7]. This scenario has driven beet armyworm resistance selection to many insecticides, including Avermectin, Pyrethroid, Organophosphate, Benzoylphenylurea, Endolsulfan, Spinosyn, and Diamide [6,7,8,9,10,11].

Pyrethroid insecticides have been used for the management of lepidopteran pests for >25 years in the US. Besides in beet armyworm, resistance to pyrethroid insecticides has been documented in heliothines, plusiines, and other *Spodoptera* species [6,12,13,14,15]. Diamide insecticides were introduced in the 2000s and are generally effective against multiple lepidopteran pests. Diamides are modulators of ryanodine receptors, a unique mode of action, with low toxicity to mammals, fish, birds, and many beneficial insects [16,17]. These properties made diamides an alternative for pest management, particularly in regions where resistance to other insecticides has evolved [18]. Diamides are classified in the mode of action group 28 by the Insecticide Resistance Action Committee (IRAC), and there are three commercial insecticides in this group: chlorantraniliprole, flubendiamide, and cyantraniliprole, released in 2008, 2009, and 2013, respectively [17]. From 2013 to 2017, the average use of diamide insecticides in Florida was approximately 9000 kg per year, with 71% of the use in vegetables and fruit crops. Previous studies reported the first chlorantraniliprole-resistant beet armyworm population reported in the US [6]. The resistant population was from the Florida Panhandle, which has a landscape with prevalent cultivation of cotton, peanut, corn, soybean, and tomato. In addition to these summer crops, winter cover crops have been more recently grown in the region acting as a source of pest infestation and promoting the year-round occurrence of beet armyworm populations in the region [6,19,20].

Fitness is the ability of an insect to adapt, survive, and reproduce in the environment [21,22,23]. Insects can show a change in fitness associated with the presence of resistance alleles [22,24]. Fitness is an important index to measure the biological changes of insect-resistant populations, since the development of resistance is often accompanied by high energy costs and adverse factors called fitness costs [22,25,26,27]. This fitness cost is a competitive disadvantage for resistant individuals when compared with a susceptible population [22,25,28]. It can be measured by investigating fitness components (i.e., survival, body weight, development time) of susceptible and resistant strains in an insecticide-free condition. The cost of resistance can be measured according to how rapidly the frequency of alleles that confer resistance decreases in confined lineages through time, in an insecticide-free environment, and without the interference of migration [22,25,29]. Thus, fitness costs can be exploited in a resistance management program because resistant insects with high fitness costs may spread slowly [29,30]. In addition, incomplete resistance, which occurs when resistant pest populations show a disadvantage from insecticide exposure relative to an insecticide-free environment, is predicted to delay the selection for resistance, due to the fitness depletion, which increases in the resistant genotypes under insecticide pressure [31,32,33].

Knowledge of the stability of resistance is also crucial for effective resistance management recommendations, considering the principle of rotation of mode of action [28,34]. In a scenario where resistance is unstable in a pest population (i.e., reduction in resistance levels in the absence of the insecticide), removing the insecticide from the spray schedule by rotation of mode of action could slow down the evolution of resistance and increase the insecticide effectiveness. How rapidly the resistance alleles decrease in confined lineages can be directly related to the fitness costs. In this scenario, there is a decrease in the resistant pest ability to survive and reproduce without the insecticide [25,35].

Because diamide insecticides were relatively recently adopted, an understanding of resistance stability and fitness costs in resistant populations is necessary to support resistance management strategies and recommendations. The objective of this study was to evaluate the fitness and stability of beet armyworm to bifenthrin and chlorantraniliprole insecticides in a US population previously reported as resistant to both insecticides. We show that the resistance to bifenthrin is complete, stable, and without fitness cost, while chlorantraniliprole resistance is unstable, likely due to fitness cost. The findings are discussed considering the implications to Insect Resistance Management (IRM) programs.

## 2. Materials and Methods

### 2.1. Beet Armyworm Colony

This study was conducted from 2018 to 2021 at the West Florida Research and Education Center (WFREC), University of Florida at Jay, FL, USA. A bifenthrin- and chlorantraniliprole-resistant (BC) colony of beet armyworms was established from a field-derived population collected from the Florida Panhandle in the 2018 crop season [6]. An insecticide susceptible colony of beet armyworms was obtained from Benzon Research Inc. (Carlisle, PA, USA).

The field-derived beet armyworm colony was reared in the entomology laboratory from F1 to F27 without further insecticide selection or further inclusion of collected insects. Beet armyworm neonates were transferred to 2.5 mL rearing cups containing a multispecies lepidopteran diet (Southland Products, Lake Village, AR, USA). At the end of the immature stage, beet armyworm pupae were transferred to metal mating cages (23 cm diameter × 30 cm height) internally lined with paper toweling as an oviposition substrate. Adults were fed with 10% honey solution, changed every two days. The eggs were collected and transferred to zip lock bags until hatching. Insect-rearing room conditions ranged from 25 ± 2 °C, 70 ± 10% relative humidity, and 14 h:10 h, light:dark photoperiod. Approximately 100 neonates were transferred to 250 mL cups containing the lepidopteran diet and reared until the appropriate instar for the insecticide bioassays. Benzon Research Inc. (Carlisle, PA, USA) provided the beet armyworm susceptible population used as a control in the bioassays because it had been kept in the laboratory for multiple generations.

### 2.2. Resistance Stability

Bifenthrin and chlorantraniliprole resistance stability in beet armyworm was examined using concentration-mortality bioassays performed with the field-derived colony at F2, F13, and F27 generations. Bioassays were also conducted using third instar larvae from insecticide susceptible (Benzon Research Inc.) colonies. Commercial formulation of Bifenthrin (Brigade 2EC; FMC Corporation, Philadelphia, PA, USA) and chlorantraniliprole (Prevathon; FMC Corporation, Newark, DE, USA) insecticides was tested at seven dilutions prepared in distilled water without any adjuvants and a control (water only). The insecticide concentration tested was defined based on label rate to control beet armyworm larvae in cotton as a starting point, and then making serial dilutions in a logarithmic (multiplicative inverse) scale. The following values were used in the bioassays as the highest concentrations: bifenthrin: 1.19 g L^−1^; chlorantraniliprole: 1.07 g L^−1^. For the conversion of field recommended rates to g L^−1^ we used a spray volume of 93.5 L ha^−1^ (10-gal acre^−1^) for the calculations. The bioassays were conducted using 4-cm leaf disks of a non-Bt cotton cultivar, DP1822XF (Monsanto, St Louis, MO, USA). The leaf-dip technique recommended by the Insecticide Resistance Action Committee (Method n. 007) was used. Leaves were removed from the middle and upper parts of the cotton plants during the vegetative stage. Leaf disks were cut and dipped individually into the insecticide solutions for 5 s with gentle agitation, allowed to dry for ≈5 min, and transferred to Petri dishes (10 cm diameter × 15 cm height). Four larvae were transferred to each leaf disk. The dishes were closed and placed in a growth chamber with the same conditions as the rearing room. The bioassays were replicated at least five times using 30–40 larvae per concentration. Larval mortality was recorded after exposure of 48 h to bifenthrin and 72 h to chlorantraniliprole. The larvae were considered dead if they did not move when prodded with a fine camelhair paintbrush.

### 2.3. Fitness of Beet Armyworm

The fitness of the field-derived and susceptible beet armyworm was determined by examination of the following fitness components: survival, development time, and body weight. As described above, one hundred neonates from the field-derived F27 generation were tested in concentration-response bioassays at seven concentrations including a control. After 48 and 72 h of exposition to bifenthrin and chlorantraniliprole insecticides, respectively, the survival was evaluated, and live larvae were individually transferred to 2.5 mL plastic cups containing multispecies lepidopteran diet. Once larval development was completed, and within 24 h after pupation, each pupa was weighed, and the sex and larval development time were recorded. A fitness index was calculated by multiplying the survival rate × pupae weight ÷ development time. The experiment was arranged in a completely randomized design with 100 larvae per insecticide concentration (one larva per replication) for each population and insecticide.

### 2.4. Statistical Analyses

Larval mortality data were analyzed using probit regression [36] in POLO PLUS v1.0 [37], adjusting for natural mortality when necessary. The susceptibility parameters estimated were the median lethal concentration (LC_50_), their respective 95% confidence limits (95% CL), and the slope and S.E. of the response curves. Resistance ratios and their respective 95% confidence intervals (CI) were determined using the susceptible population as references for comparison [38]. Differences in fitness (i.e., body weight and development time) among laboratory and field populations were compared using one-way analysis of variance and *t*-test. Differences caused by concentrations within insecticides were compared using one-way analysis of variance and means were separated by Tukey’s honest significant difference test. The significance level was set at 0.05 for all tests (SAS version 9.4, SAS Institute, Cary, NC, USA).

## 3. Results

### 3.1. Resistance Stability

The field-derived beet armyworm showed LC_50_ (g/L) of 3310 (2085–15,520) for the F2 generation and >250 for the F13 and F27 when exposed to bifenthrin in concentration-mortality bioassays. The susceptible population exhibited consistently high susceptibility to bifenthrin with LC_50_ from 0.12 (0.03–0.27) to 0.32 (0.16–0.53) g/L (Table 1). The resistance ratio for all generations exposed to bifenthrin was >1000.

Chlorantraniliprole bioassays performed with the field-derived colony at F2, F13, and F27 generations demonstrated a decrease in LC_50_ values (Table 2). The LC_50_ (g/L) for the F2 generation was >139.92, declining to 121.93 (32.00–273.00) and 15.92 (1.79–81.81) for the F13 and F27, respectively. The LC_50_ (g/L) of the susceptible population was low and fluctuated from 1.02 (0.19–2.79) to 2.12 (1.31–3.46). The resistance ratio among field and susceptible beet armyworm populations in the absence of selection pressure to chlorantraniliprole insecticide was 630, 80, and 15.

### 3.2. Fitness of Beet Armyworm

Field-derived F27 beet armyworm larval survival was reduced only at high concentrations of chlorantraniliprole, but not at any bifenthrin concentration (Figure 1). Both chlorantraniliprole and bifenthrin reduced the larval survival of susceptible beet armyworm (Figure 1). Susceptible beet armyworm shows longer larval development time (neonate–pupae) on both insecticides and control compared to the field-derived F27 population (Figure 2a,b). The field-derived beet armyworm had a longer pupal development time and heavier pupae than susceptible beet armyworm when exposed to chlorantraniliprole, but these parameters were similar for the field-derived and susceptible beet armyworm exposed to bifenthrin (Figure 2c,f). Susceptible and field-derived beet armyworm had a lower fitness index with the increase in chlorantraniliprole concentration (Figure 3). In contrast, field beet armyworm did not have fitness compromised by any concentration of bifenthrin (Figure 3).

## 4. Discussion

Our results investigating beet armyworm resistance stability have indicated no decrease in bifenthrin resistance for at least a three-year period (i.e., 27 generations) when insecticide exposure, and consequent selection pressure for resistance, was suspended. However, susceptibility to chlorantraniliprole dropped approximately 160-fold through this three-year period. Our results indicate that beet armyworm resistance to bifenthrin is stable, but unstable to chlorantraniliprole.

Unstable resistance can be successfully managed at field level by switching off the selection pressure with replacement of the insecticide other than a pyrethroid (i.e., mode of action rotation). In the case of bifenthrin resistance, the susceptibility of resistant individuals may not change in the absence of selection pressure (stable resistance) [23,28,35,39]. The results reported in the present study with beet armyworms indicate that rotation of mode of action of insecticides should not be considered a “one-size-fits-all” IRM recommendation. The monitoring and study of the stability of resistance to various insecticides used for target pests when designing an IRM program is a critical point.

A landscape effect of migratory pest populations should be considered when studying the stability of resistance. In the present study, the stability of resistance was examined in the absence of migration of susceptible individuals [28,40]. However, if the primary source of pest migration is from adjacent unsprayed host crops, this pest movement in the landscape would be expected to enhance the rate of resistance reversion. If the source of migration was from nearby sprayed crops, the resistance reversion would be expected to occur more slowly.

In addition, multiple factors may affect the stability of resistance in a laboratory-reared insect population, including fitness costs, mutation, selection, and genetic drift [23,35,39,40,41]. The stable bifenthrin resistance might be a result of the absence of fitness costs expressed by resistant strains. However, fitness costs might be environmentally dependent and may not occur in laboratory conditions. Second, bifenthrin resistance alleles may have been near fixation in a very homogeny field population leading to a low increase in heterozygosity [42]. The simplest explanation for the instability of chlorantraniliprole resistance reported in this study is that beet armyworms had a fitness cost when the resistant population was collected in 2018 [6]. This fitness cost may have decreased over the three years in the laboratory along with the reduction in the frequency of chlorantraniliprole-resistant alleles [43]. This hypothesis is consistent with the concept that the fitness cost of resistance favors restoration of susceptibility in the absence of exposure to insecticide [41,44]. Because the fitness study was performed when beet armyworm was no longer resistant to chlorantraniliprole, any fitness cost accompanied by chlorantraniliprole resistance was no longer evident. Future studies should explore whether the new field-resistant populations of beet armyworm show fitness cost to chlorantraniliprole.

The cause of the fitness cost depends entirely on whether the resistance is metabolic (a quantitative trait) or target-site resistance (a discrete trait). In the case of metabolic resistance, the cause of the fitness cost is a straightforward trade-off between resource allocation into xenobiotic metabolism or allocation into nutrition metabolism [45]. *Culex pipiens* mosquitoes that over-express esterases were shown to carry an average of 30% fewer lipids, glycogen, and glucose than their wild-type counterparts [46]. In the case of target-site resistance, the cost is less predictable. Xenobiotics often take effect by altering the function of a constitutively expressed (‘lethal’) gene or protein. Target-site resistance modifications are themselves associated with fitness effects, due to potential alterations to the function of the protein itself, and the biochemistry surrounding that. If the functional effects of the mutation are significant, then the cost of resistance will be high [28,45].

This study performed with a beet armyworm population showed resistance to two distinct insecticides commercially launched over 60 years apart. Due to the different mode of action and lack of cross-resistance between them, beet armyworm potentially developed distinct resistance mechanisms for each insecticide [47]. These distinct resistance mechanisms also explain the stability of resistance of beet armyworm to bifenthrin and unstable resistance to chlorantraniliprole. The resistance to diamide in beet armyworm is likely to happen due to a mutation on the ryanodine receptor, detoxification metabolism, or decrease target binding affinity [11,47,48,49,50]. Generally, pyrethroid resistance is grounded on the increased detoxification abilities because of over-expression of p450s, esterases, and glutathione S-transferases, and reduced sensitivity of the insecticide target proteins on the voltage-gated sodium channel [6,12,51,52]. Further studies on the biochemical mechanism of this resistance in beet armyworms should be considered to provide more detailed genetic evidence to support these assumptions.

Reduction in fitness was not observed when beet armyworm was exposed to bifenthrin compared to the control. This finding suggests complete resistance, which is an undesirable condition for beet armyworm management. In this situation, the fitness of resistant individuals is equal or better on the insecticide than in the insecticide-free environment [31]. Unlike incomplete resistance, complete resistance is difficult to reverse and may speed the spread of resistance alleles, explaining why beet armyworm bifenthrin resistance is fixed in populations and present across regions [31,53]. Effective IRM programs depend upon the possible types of resistance mechanisms and factors that enhance susceptibility recovery, sometimes neglected when designing insect resistance management strategies. The common recommendation of rotation of mode of action is in part based on the possibility of susceptibility recovery of pest populations. Understanding fitness cost and resistance stability are necessary to prevent the spread of resistance alleles by devising an appropriate resistance management strategy [22,23,35]. Studies to enhance susceptibility recovery should be performed with target pests, preferably before the adoption of a specific technology. In addition to rotating insecticide mode of action, combining the use of insecticide mixtures or alternations of different insecticides with other IPM tactics should be considered, such as the use of biological and cultural methods, conservation of beneficial insects, and the use of selective insecticides.

Although the beet armyworm used as a susceptible (laboratory) population does not share the same genetic background of the field-derived population, the differences in fitness components (i.e., life-history traits) documented in this study under insecticide exposure likely represent the insecticide effect rather than population discrepancies. In conclusion, the combined findings of no significant declines in resistance in the absence of selection and no fitness differences between resistant and susceptible insects suggest that bifenthrin resistance in beet armyworm is stable, complete, and without fitness cost. Resistance to chlorantraniliprole in beet armyworm appears to be unstable, likely due to fitness cost. These findings advance our understanding of insecticide resistance and its stability in beet armyworm and provide scientific evidence and information for the improvement of insect pest resistance management.

## Figures and Tables

**Figure 1 insects-13-00365-f001:**
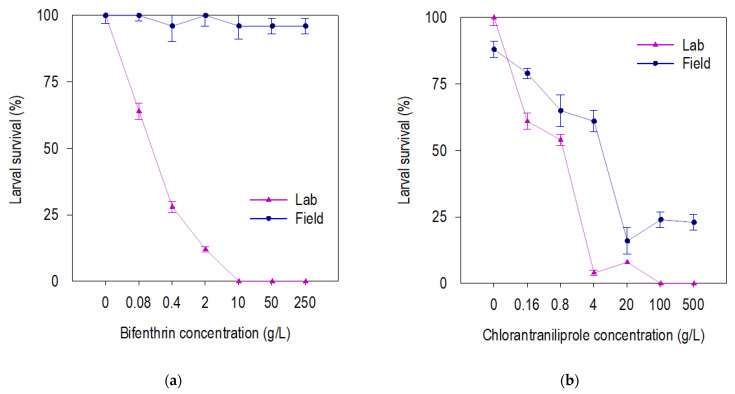
Larval survival (mean percentage ± SEM) of beet armyworm field-derived and laboratory populations after seven days of exposure to bifenthrin (**a**) and chlorantraniliprole (**b**) insecticides in concentration-response bioassay.

**Figure 2 insects-13-00365-f002:**
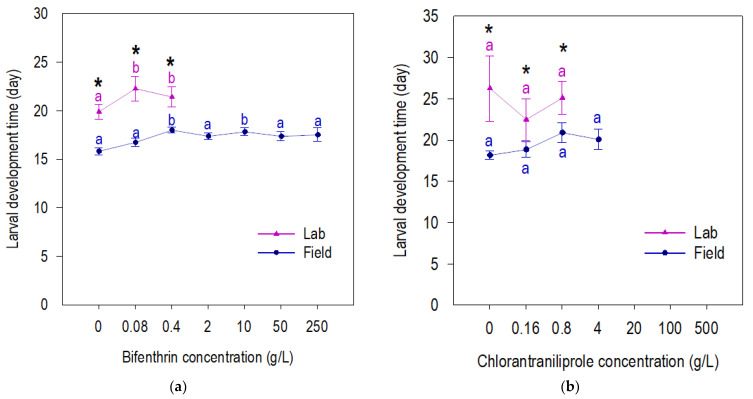
Fitness components (larval development time (**a**,**b**), pupal development time (**c**,**d**), pupal weight (**e**,**f**)) of beet armyworm field-derived and laboratory populations exposed to bifenthrin (**left**) and chlorantraniliprole (**right**) insecticides in concentration-response bioassay. Means accompanied by the same letter within populations (lab and field) are not significantly different (*p* > 0.05; Tukey’s HSD). * indicates significative difference among laboratory and field beet armyworm for a specific concentration, *p* < 0.05.

**Figure 3 insects-13-00365-f003:**
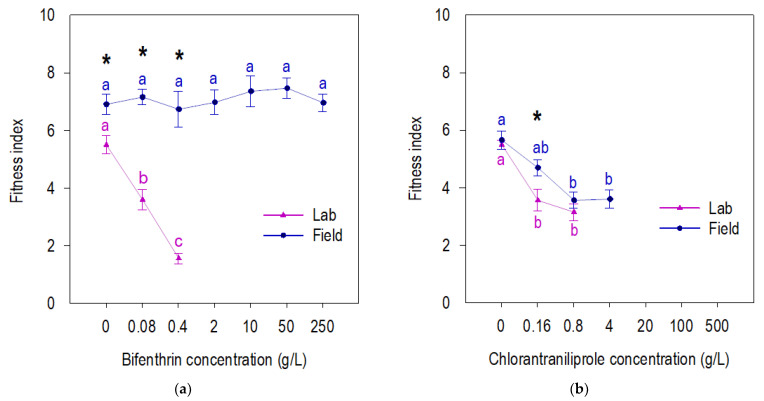
Fitness index (survival rate × pupa weight ÷ development time) of beet armyworm field-derived and laboratory populations exposed to bifenthrin (**a**) and chlorantraniliprole (**b**) insecticides in concentration-response bioassay. Means accompanied by the same letter within populations (lab and field) are not significantly different (*p* > 0.05; Tukey’s HSD). * indicates significative difference among laboratory and field beet armyworm for a specific concentration, *p* < 0.05.

**Table 1 insects-13-00365-t001:** Resistance stability of beet armyworm to bifenthrin (pyrethroid) insecticide. Susceptibility of field-derived and laboratory beet armyworm populations was tested in concentration-response bioassay using third-instar larvae (IRAC method 007).

Population	Generation (Year)	N ^a^	Equation	χ^2^	*p* ^c^	LC50 (95% CL) ^b^	RR (95% CL) ^d^
Field	F2 (2018)	300	y = −7.28 + 1.76x	2.04	0.56	3310.00 (2085–15,520)	10,071 (4426–22,916)
	F13(2019)	200	y = −0.70 + −0.08x	nc ^e^	nc	>250.00	>2083
	F27(2020)	148	y = −2.07 + 0.16x	nc	nc	>250.00	>1666
Laboratory	F2 (2018)	300	y = −0.26 + 1.78x	0.02	0.99	0.32 (0.16–0.53)	1
	F13(2019)	250	y = −0.98 + 1.10x	1.79	0.77	0.12 (0.03–0.27)	1
	F27(2020)	150	y = 1.00 + 1.24x	0.84	0.42	0.15 (0.06–0.27)	1

^a^ N, number of individuals tested. ^b^ LC_50_, lethal concentration to cause mortality in 50% of individuals expressed in g/L and 95% confidence limits (95% CL). ^c^
*p*-value associated with the chi-square, goodness-of-fit test. ^d^ RR, resistance ratio and 95% confidence limits (95% CL). RR values are considered significant (relative to the respective laboratory population) if the 95% CL does not include 1. ^e^ nc, not calculated due to lack of mortality even at the highest concentration tested.

**Table 2 insects-13-00365-t002:** Resistance stability of beet armyworm to chlorantraniliprole (diamide) insecticide. Susceptibility of field-derived and laboratory beet armyworm populations was tested in concentration-response bioassay using third-instar larvae (IRAC method 007).

Population	Generation (Year)	N ^a^	Equation	χ^2^	*p* ^c^	LC_50_ (95% CL) ^b^	RR (95% CL) ^d^
Field	F2 (2018)	144	y = −3.06 + 0.07x	nc ^e^	nc	>139.92	629 (13–22,215)
	F13(2019)	200	y = −1.80 + 0.86x	2.81	0.42	121.93 (32.00–273.00)	80 (25–252)
	F27(2020)	127	y = −0.64 + 0.53x	0.87	0.97	15.92 (1.79–81.81)	15 (1–123)
Laboratory	F2 (2018)	300	y = −0.39 + 1.19x	2.29	0.51	2.12 (1.31–3.46)	1
	F13(2019)	250	y = −0.16 + 0.92x	3.05	0.21	1.51 (0.11–8.18)	1
	F27(2020)	122	y = −0.09 + 1.07x	1.69	0.42	1.02 (0.19–2.79)	1

^a^ N, number of individuals tested. ^b^ LC_50_, lethal concentration to cause mortality in 50% of individuals expressed in g/L and 95% confidence limits (95% CL). ^c^
*p*-value associated with the chi-square, goodness-of-fit test. ^d^ RR, resistance ratio and 95% confidence limits (95% CL). RR values are considered significant (relative to the respective laboratory population) if the 95% CL does not include 1. ^e^ nc, not calculated due to lack of mortality even at the highest concentration tested.

## Data Availability

The data presented in this study are available on request from the corresponding author.

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
