# Peer review of "Spodoptera exigua (Hubner) (Lepidoptera: Noctuidae) Fitness and Resistance Stability to Diamide and Pyrethroid Insecticides in the United States"

_insects, 2022, doi:10.3390/insects13040365_

Round 1

Reviewer 1 Report

In the present study, the authors evaluated beet armyworm fitness and resistance stability to diamide and pyrethroid insecticides in USA. The paper generally presents an interesting topic that can be of value for publication in “Insects” because the knowledge related to fitness and stability of insecticide resistance of insect pests in a particular region may help the community to choose an alternative insecticide with different mode of action to minimize the damage.

However, there are some concerns needed to be clarify before publication.

  1. In 2.2 Resistance stability, the concentrations of insecticides used in the experiments were not mentioned and should be clearly presented.
  2. In Table 1 and Table 2, only slope is presented and constant is not presented for the probit regression model. The constant should be presented.
  3. In Table 1, LC50 of bifenthrin to beet armyworm at F13 and F27 (>250 g/L) was not calculated due to the lack of mortality even at the highest concentration but LC50 of 3310 g/L at F2 was calculated, which is much higher than 250 g/L. Therefore, LC50 of bifenthrin to beet armyworm at F13 and F27 should be obtained.
  4. Please explain how to get the LC50 of chlorantraniliprole to beet armyworm at F2 of >139.92 in Table 2, and how to obtain the RR of 629 (13 – 22215) when LC50 was not precisely calculated.
  5. The equation of fitness index of survival rate × pupa weight ÷ development time in the Results should be moved to the Materials and Methods. If there is a reference, please cite it in the text.
  6. In the Results, the values are not matching between text and table. Please pay attention to the precision of number.
  7. Please discuss the reason that the bifenthrin resistance was stable in beet armyworm with references to support that the other pyrethroid insecticides with the same mode of action with bifenthrin showed stable resistance in insects.
  8. All scientific names of organisms and names of variables should be italic in the text.
  9. In Line 61, please provide the complete name for IRAC.
  10. In Line 171, “Differences concentrations” should be replaced with “Differences caused by concentrations”.

Author Response

Author's Response

March 2022

I am submitting the revised version of the research article "Spodoptera exigua (Hubner) (Lepidoptera: Noctuidae) fitness and resistance stability to diamide and pyrethroid insecticides in the United States".

We appreciate your comments along with the reviewer/suggestions and constructive criticisms. Please find attached our detailed point-by-point responses to the reviewer's suggestions and the revised manuscript.

Sincerely,

Marcelo Rabelo

University of Florida

Comments and Suggestions for Authors

In the present study, the authors evaluated beet armyworm fitness and resistance stability to diamide and pyrethroid insecticides in USA. The paper generally presents an interesting topic that can be of value for publication in “Insects” because the knowledge related to fitness and stability of insecticide resistance of insect pests in a particular region may help the community to choose an alternative insecticide with different mode of action to minimize the damage.

However, there are some concerns needed to be clarify before publication.

1. In 2.2 Resistance stability, the concentrations of insecticides used in the experiments were not mentioned and should be clearly presented.

We have added the label rate for each insecticide used as the highest concentration in each bioassay.

2. In Table 1 and Table 2, only slope is presented and constant is not presented for the probit regression model. The constant should be presented.

The authors found that reporting a constant from the Probit regression along with the LC 50, X2, and P values is overlaying and distracting. The LC50 and RR are the most important points in the table, and it is already validated by the X2 and P.

3. In Table 1, LC50 of bifenthrin to beet armyworm at F13 and F27 (>250 g/L) was not calculated due to the lack of mortality even at the highest concentration but LC50 of 3310 g/L at F2 was calculated, which is much higher than 250 g/L. Therefore, LC50 of bifenthrin to beet armyworm at F13 and F27 should be obtained.

Each LC50 was estimated in a different year/bioassay. The range of concentration was always based on two factors, label recommended rate (14-27 fl oz/acre) and the change in susceptibility of the pest. Even with the standardized procedure and dose, the insect mortality oscillates and so does the estimation on the polo plus. At the F13 and F27, no estimation was possible.

4. Please explain how to get the LC50 of chlorantraniliprole to beet armyworm at F2 of >139.92 in Table 2, and how to obtain the RR of 629 (13 – 22215) when LC50 was not precisely calculated.

Even though the LC50 for the resistant population was not estimated, This RR estimation was automatically computed by the polo-plus. It is more reliable than simply dividing 139.92 / 2.12 LC50/LC50 because it contains confidential limits. Otherwise, we would use >66 with no confidence limits a much vaguer estimation.

5. The equation of fitness index of survival rate × pupa weight ÷ development time in the Results should be moved to the Materials and Methods. If there is a reference, please cite it in the text.

The equation was removed from Results appears only in Materials and Methods

6. In the Results, the values are not matching between text and table. Please pay attention to the precision of number. Updated. As it reads

Line 183: LC50 from 0.12 (0.03 – 0.27) to 0.32 (0.16–0.53) g/L (Table 1).  

Line 186-187: The LC50 (g/L) for the F2 generation was > 139.92, declining to 121.93 (32.00 – 273.00) and 15.92 (1.79 – 81.81) for the F13 and F27, respectively.

7. Please discuss the reason that the bifenthrin resistance was stable in beet armyworm with references to support that the other pyrethroid insecticides with the same mode of action with bifenthrin showed stable resistance in insects.

We have discussed the fact that the stability of resistance can be environmentally dependent and may not occur under lab conditions or the resistance may have been near fixation, with a very slow increase in heterozygosity.

This is supported by: Ahmad, M., Sayyed, A.H., Crickmore, N. and Saleem, M.A., 2007. Genetics and mechanism of resistance to deltamethrin in a field population of Spodoptera litura (Lepidoptera: Noctuidae). Pest Management Science: formerly Pesticide Science63(10), pp.1002-1010.

8. All scientific names of organisms and names of variables should be italic in the text. Updated

9. In Line 61, please provide the complete name for IRAC. Updated

10. In Line 171, “Differences concentrations” should be replaced with “Differences caused by concentrations”. updated

Reviewer 2 Report

General comments

  1. The title may not be proper. In this manuscript, only one bifenthrin- and chloranthraniliprole-resistant strain collected from a field from the Florida Panhandle was used. Could this strain reflect the situation in the United States like in the title?
  2. This manuscript studied the susceptibility of the bifenthrin- and chloranthraniliprole-resistant strain to those two insecticides for three years without exposure to any insecticide. In abstract and results, the data were collected from F2, F13 and F27 generations. However, in materials and methods (line 130) F1, F13 and F27 were described. Pleases clarify it, F1 or F2.
  3. The resistance ratio to chloranthraniliprole at F13 and F27 were shown 81 and 16 in the abstract, respectively, but 80 and 15 in the results and Table 2, respectively. Also the RR value at F2 appeared in the Table 2 (629) is different from those in the abstract and results (630). Please clarify it.
  4. P. 3, line 104, Is IRM programs the Integrated Risk Management?
  5. In figure 2 and figure 3 legends, “…are not significantly different (P < 0.05;…” “* indicate significative difference among laboratory and field beet armyworm for a specific concentration, P > 0.05.”   If there is no significant difference, P value should be > 0.05. If there is significant difference, P value should be < 0.05.
  6. P. 9, lines 268-269, “Because the fitness study was performed when beet armyworm was no longer resistant to chlorantraniliprole,”   The RR value is 15 at F27 in Table 2. It should still be resistant.

Editing comments

  1. All scientific names have to be italic.
  2. P. 2, line 65, is (Wieben 2019) a citation? Using the number format.
  3. P. 10, line 328, “writing—Rabelo, X.X.;” What does X.X. mean here?
  4. Number 6 and 45 references are the same.
  5. P. 5, line 212, (Figure 2a, b); line 216, (Figure 2c-f)

Author Response

Author's Response

March 2022

I am submitting the revised version of the research article "Spodoptera exigua (Hubner) (Lepidoptera: Noctuidae) fitness and resistance stability to diamide and pyrethroid insecticides in the United States".

We appreciate your comments along with the reviewer/suggestions and constructive criticisms. Please find attached our detailed point-by-point responses to the reviewer's suggestions and the revised manuscript.

Sincerely,

Marcelo Rabelo

University of Florida

Comments and Suggestions for Authors

General comments

1. The title may not be proper. In this manuscript, only one bifenthrin- and chloranthraniliprole-resistant strain collected from a field from the Florida Panhandle was used. Could this strain reflect the situation in the United States like in the title?

This is a good point – The majority of the resistance studies in BAW populations were performed in China – This is why we highlighted the US up front. Also, this is the first resistance study in chlorantraniliprole in BAW in the US.

2. This manuscript studied the susceptibility of the bifenthrin- and chloranthraniliprole-resistant strain to those two insecticides for three years without exposure to any insecticide. In abstract and results, the data were collected from F2, F13 and F27 generations. However, in materials and methods (line 130) F1, F13 and F27 were described. Pleases clarify it, F1 or F2.

The correct generation is F2 - updated

3. The resistance ratio to chloranthraniliprole at F13 and F27 were shown 81 and 16 in the abstract, respectively, but 80 and 15 in the results and Table 2, respectively. Also the RR value at F2 appeared in the Table 2 (629) is different from those in the abstract and results (630). Please clarify it.

The values in the abstract were round up. However, we have corrected to the original values presented in the tables to avoid confusion.

4. P. 3, line 104, Is IRM programs the Integrated Risk Management?

Insect resistance management - updated

5. In figure 2 and figure 3 legends, “…are not significantly different (P < 0.05;…” “* indicate significative difference among laboratory and field beet armyworm for a specific concentration, P > 0.05.”   If there is no significant difference, P value should be > 0.05. If there is significant difference, P value should be < 0.05.

Correct- updated

6. P. 9, lines 268-269, “Because the fitness study was performed when beet armyworm was no longer resistant to chlorantraniliprole,”   The RR value is 15 at F27 in Table 2. It should still be resistant.

    The RR was 15. However, the 95% confidence limits reges from 1 - 123. As stated on the tables foot note - d RR values are considered significant (relative to the respective laboratory population) if the 95% CL does not include 1.

Editing comments

  1. All scientific names have to be italic. updated
  2. P. 2, line 65, is (Wieben 2019) a citation? Using the number format. updated
  3. P. 10, line 328, “writing—Rabelo, X.X.;” What does X.X. mean here? updated
  4. Number 6 and 45 references are the same. updated
  5. P. 5, line 212, (Figure 2a, b); line 216, (Figure 2c-f) - updated

Reviewer 3 Report

I am unable to review this paper because the growth metrics in Figures 1-3 comparing field to lab results are not readable.  The only difference between the plotted data for these two treatments is color.  The authors did not use different symbol shapes to distinguish field from lab results.  Since I am partially color blind like many people with blue eyes, I do not see any color differences and not even sure if there were color differences provided.  Because of this, I can not review any of the growth metric and fitness data.  So no point of writing a review.

Author Response

Author's Response

March 2022

I am submitting the revised version of the research article "Spodoptera exigua (Hubner) (Lepidoptera: Noctuidae) fitness and resistance stability to diamide and pyrethroid insecticides in the United States".

We appreciate your comments along with the reviewer/suggestions and constructive criticisms. Please find attached our detailed point-by-point responses to the reviewer's suggestions and the revised manuscript.

Sincerely,

Marcelo Rabelo

University of Florida

Comments and Suggestions for Authors

I am unable to review this paper because the growth metrics in Figures 1-3 comparing field to lab results are not readable.  The only difference between the plotted data for these two treatments is color.  The authors did not use different symbol shapes to distinguish field from lab results.  Since I am partially color blind like many people with blue eyes, I do not see any color differences and not even sure if there were color differences provided.  Because of this, I can not review any of the growth metric and fitness data.  So no point of writing a review.

Reply: We have updated the figures with different symbol shapes to distinguish the field from lab results.